# FLOWSTAR-Energy: a high resolution wind farm wake model

Amy Stidworthy<sup>1</sup>, David Carruthers<sup>1</sup> <sup>1</sup>CERC Ltd, 3 Kings Parade, Cambridge, CB2 1SJ, UK *Correspondence to*: Amy Stidworthy (amy.stidworthy@cerc.co.uk)

- Abstract. A new model, FLOWSTAR-Energy, has been developed for the practical calculation of wind farm energy production. It includes a semi-analytic model for airflow over complex surfaces (FLOWSTAR) and a wind turbine wake model that simulates wake-wake interaction by exploiting some similarities between the decay of a wind turbine wake and the dispersion of plume of passive gas emitted from an elevated source. Additional turbulence due to the wind shear at the wake edge is included and the assumption is made that wind turbines are only affected by wakes from upstream wind
- turbines. The model takes account of the structure of the atmospheric boundary layer, which means that the effect of atmospheric stability is included. A marine boundary layer scheme is also included to enable offshore as well as onshore sites to be modelled.

FLOWSTAR-Energy has been used to model three different wind farms and the predicted energy output compared with measured data. Maps of wind speed and turbulence have also been calculated for two of the wind farms. The Tjaæreborg

- wind farm is an onshore site consisting of a single 2 MW wind turbine, the NoordZee offshore wind farm consists of 36 V90 VESTAS 3 MW turbines and the Nysted offshore wind farm consists of 72 Bonus 2.3 MW turbines. The NoordZee and Nysted measurement datasets include stability distribution data, which was included in the modelling. Of the two offshore wind farm datasets, the NoordZee dataset focuses on a single 5-degree wind direction sector and therefore only represents a limited number of measurements (1,284); whereas the Nysted dataset captures data for seven 5-degree wind direction sectors
- and represents a larger number of measurements (84,363). The best agreement between modelled and measured data was obtained with the Nysted dataset, with high correlation (0.98 or above) and low normalised mean square error (0.007 or below) for all three flow cases. The results from Tjæreborg show that the model replicates the Gaussian shape of the wake deficit two turbine diameters downstream of the turbine, but the lack of stability information in this dataset makes it difficult to draw conclusions about model performance.
- One of the key strengths of FLOWSTAR-Energy is its ability to model the effects of complex terrain on the airflow. However, although the airflow model has been previously compared extensively with flow data, it has so far not been used in detail to predict energy yields from wind farms in complex terrain. This will be the subject of a further validation study for FLOWSTAR-Energy.

#### **1** Introduction

Wind turbines generate electricity by extracting energy from the wind; this process creates a 'wake' downwind of the turbine. When wind turbines are sited individually or far apart, the effects of wind turbine wakes on the energy generation of other turbines is insignificant. However, wind turbines are often grouped together into wind 'farms' for greater efficiencies in terms of land use, operation and maintenance. Wake effects within a wind farm can cause significant power losses; up to 40% in some cases (e.g. Barthelmie et al., 2009). The greatest wake losses occur where the wind speed is below the 'rated

- wind speed' of a turbine, typically 13 or 14 ms<sup>-1</sup> at hub height, as is often the case. Below the rated wind speed, the energy a turbine extracts from the wind increases sharply with increasing wind speed; above the rated wind speed, blade pitch control mechanisms keep the turbine power output constant at the turbine's rated power level. Therefore, below the rated wind
- speed, any reduction in wind speed, for example due to the wake from an upstream turbine, has a significant impact on power output. This loss of efficiency and potential for variability in wind farm energy production can be a serious problem for energy distribution authorities who operate complex energy networks. Therefore, it is important to account for wake effects both at the design stage of a wind farm and when predicting power output during operation. In the study of wind turbine wakes, the 'near wake' is the term given to the part of the wake that is up to approximately two
- turbine diameters downstream from the turbine; here, root and tip vortices generated by the turbine blades create a tubular shear layer. Beyond the complex near wake region is the part of the wake known as the 'far wake', where the wake can be characterised more simply as a region of reduced axial wind speed and increased turbulence levels (e.g. Sanderse et al., 2011). Over the past twenty years, models have been developed to study the aerodynamics of wind turbines and the behaviour of wind turbine wakes with varying levels of complexity, as reviewed in Sanderse et al., 2011. Simple kinematic
- models only simulate the far wake; increasingly the trend is towards more complex computational fluid dynamics (CFD) methods that simulate the whole wake, including the complex near wake region. These models offer great insights into the behaviour of wind turbine wakes and are important for turbine design calculations, but require extensive computational resources and are not yet a practical tool for predicting wind farm energy production.
- In this paper we describe a new model, FLOWSTAR-Energy, which has been developed for the practical calculation of wind farm energy production. This includes a model for airflow and turbulence over complex terrain and a novel plume-based method for the downstream development of turbine wakes. FLOWSTAR-Energy is described in Section Error! Reference source not found. and we present validation of the wake model for three sites in Section Error! Reference source not found.

#### 2 Model description

FLOWSTAR-Energy comprises a semi-analytic model for airflow over complex surfaces (FLOWSTAR) together with a wind turbine wake model including wake-wake interaction and impacts on the mean flow. FLOWSTAR has previously been described (Carruthers et al., 1988) and validated (Carruthers et al., 2014, Stocker et al., 2016, CERC, 2016 - Askervein Hill

and Blashaval) and has been subject to extensive use as a component part of the Atmospheric Dispersion Modelling System (ADMS) (CERC, 2015). In essence it represents the lower atmosphere with three layers: a boundary layer; a capping inversion; and the free troposphere. It uses linearised flow equations together with a non-linear surface boundary condition. Near the surface the impact of shear stress perturbations on the mean flow is represented using a mixing length closure. The model takes account of spatial variations in both terrain elevation and surface roughness. The solution is continuous in the

- vertical and has a resolution in the horizontal direction as fine as the terrain data and/or computational resources allow. The wind turbine wake model exploits some similarities between the decay of the wake behind a wind turbine and the dispersion of a plume of passive gas emitted from an elevated source: both are advected downstream and mix with ambient air, which in the case of the wake decreases its intensity and in the case of the plume decreases the concentration of gas in
- the plume. The method is thus to represent the fully-expanded wind turbine wake immediately downwind of a wind turbine by a cuboid-shaped 'volume source' that passively 'emits' the wind speed deficit. The model uses 1-D momentum theory to determine the maximum wind speed deficit in the far wake from turbine thrust coefficient data, which is used to determine the volume source strength and dimensions. The wind speed deficit is the quantity which is then 'dispersed' downwind. This dispersion calculation is conducted using ADMS, modified to include the additional shear-induced turbulence in the wake.
- The overall wake within and downstream of groups of turbines (i.e. a wind farm) is modelled by considering the wake effects from individual wind turbines in downstream order, so that wakes from upstream wind turbines affect the flow field used when characterising the downstream wind turbine volume sources and when dispersing their wakes.

#### 2.1 Representation of a wind turbine as an effective volume source

Figure 1 shows a schematic diagram of the expanding stream-tube of flow through a turbine together with the effective volume source used in the model.

According to classical 1-D momentum theory (e.g. Hansen, 2008), the maximum wind speed deficit in a fully expanded wind turbine wake is equal to 2aU and the maximum wind speed deficit in the partially expanded wake at the turbine is equal to aU, where U is the upstream wind speed at hub height in units of ms<sup>-1</sup> and a is known as the 'axial induction factor'. Following 0, the axial induction factor a is related to the thrust coefficient  $C_T$  as follows:

$$C_T = \begin{cases} 4a(1-a), & a \le a_c \\ 4(a_c^2 + (1-2a_c)a), & a > a_c \end{cases}$$
 (1)

| where $a_c \approx 0.2$ . Thus, Eq. | (1) can be used to write |
|-------------------------------------|--------------------------|

an expression for *a* as a function of  $C_T$ :

$$a = \begin{cases} \frac{1}{2} \left( 1 - \sqrt{1 - C_T} \right) & C_T \le 0.64 \\ \frac{C_T - 4a_c^2}{4(1 - 2a_c)} & C_T > 0.64 \end{cases}.$$
(2)

Conservation of mass in the expanding stream-tube leads to a relationship between the diameter of the fully expanded wake  $D_W$  and the diameter of the turbine rotor *D*:

$$D_W = D_{\sqrt{\frac{(1-a)}{(1-2a)}}}.$$
(3)

For pragmatic reasons the initial cross-section of the volume source is taken as square rather than circular, but this has little 5 impact on the developing wake. The dimensions of the source are calculated so that the crosswind area is equal to that of the fully expanded wake, hence:

$$dy = dz = \frac{D}{2} \sqrt{\frac{\pi(1-a)}{(1-2a)}},$$
(4)

where dy is the horizontal crosswind extent of the volume source and dz is the depth of the volume source. The along-wind extent of the volume source is set to ten percent of the crosswind extent dy. Since modern wind turbines have quick and

10 efficient mechanisms to bring the wind turbine rotor perpendicular to the inflow wind, the model assumes zero yaw, therefore the volume source is aligned perpendicular to the upstream wind.

The volume source strength is calculated using concentration as a surrogate for the wind speed deficit. The maximum wind speed deficit  $\Delta U_{max}$  can be expressed as

$$\Delta U_{max} = \frac{QV_{src}}{\dot{\gamma}},\tag{5}$$

where *Q* is the source strength (in units of ms<sup>-2</sup>),  $V_{src}$  is the source volume, and  $\dot{V}$  is the volume flow rate through the source; therefore,

$$Q = \frac{2aU^2}{0.1dy}.$$
(6)

Note that the volume source produces initial 'top-hat' vertical and horizontal profiles of wind speed deficit, which decay to Gaussian-shape profiles as they evolve downstream.

# 20 2.2 'Dispersion' of the 'source'

The effective volume source representing the initial wind speed deficit in the wake is decomposed into a maximum of ten thin crosswind source elements, where the width of each source element is a function of the proximity of the element to the receptor, while the difference in streamwise distance between source elements and the receptor is constrained not to vary too rapidly. The wake deficit at each receptor is then calculated by summing the contributions from each element.

The wake deficit contribution  $\overline{C}(x, y, z)$  calculated for each element from a crosswind vertical slice of length  $L_s$  and height  $L_1$  is given by Eq. (7).

5

$$\bar{\bar{C}}(x,y,z) = \frac{\bar{Q}_s}{4U} \left[ \operatorname{erf}\left(\frac{y + \frac{1}{2}L_s}{\sigma_y \sqrt{2}}\right) - \operatorname{erf}\left(\frac{y - \frac{1}{2}L_s}{\sigma_y \sqrt{2}}\right) \right] \times \left[ \operatorname{erf}\left(\frac{z + \frac{1}{2}L_1 - z_s}{\sigma_z \sqrt{2}}\right) - \operatorname{erf}\left(\frac{z - \frac{1}{2}L_1 - z_s}{\sigma_z \sqrt{2}}\right) \right] + \operatorname{reflection terms}$$
(7)

The source strength  $\overline{\bar{Q}}_s$  is in units of ms<sup>-2</sup>.

Field experiments and research have shown that the dispersion parameters  $\sigma_y$  and  $\sigma_z$  vary with downwind distance x from a source of airborne emitted material in a way that depends on: the atmospheric boundary layer height (*h*); the Monin Obukhov length ( $L_{M0}$ ), which is a measure of atmospheric stability; height of the source ( $z_s$ ); and the height of the plume as it travels downwind. For reviews of this subject, see Hunt et al., 1988a, Hanna and Paine, 1989 and Weil, 1985. The approach adopted is to use formulae that have been developed and broadly accepted for specific ranges of the parameters  $z_s/h$  (source height),  $h/L_{M0}$  (stability) and x/h (downwind distance). Interpolation formulae have then been constructed to cover the complete parameter range. The basis for these formulae is set out at length in Hunt et al., 1988a.

#### 10 2.3 Shear-induced turbulence

At the edge of a turbine wake there is a gradient in the speed of the air flow, which generates additional turbulence. Based on Bevilaqua and Lykoudis, 1978, an extra component of turbulence,  $\sigma_{shear}$ , has been included as follows:

$$\sigma_{shear}\left[i+1\right] = \begin{cases} 0.4|\Delta U|\frac{x}{X_{crit}}, & x \le X_{crit} \\ \sigma_{shear}\left[i\right] \times e^{-dt/t}, & x > X_{crit} \end{cases},$$
(8)

where x is the downwind distance from the effective volume source and  $|\Delta U|$  is the local wind speed deficit. Here, *i* and 15 *i* + 1 represent consecutive calculation points in a downstream direction along the wake centreline; *dt* represents the time taken by the wake centreline to travel from the *i*th to the (*i* + 1)th point. *t* represents the time taken by the plume to spread the width (depth) of the wake:

$$t = \frac{2R}{\sigma_{tot}}.$$
(9)

*R* is the wake radius, defined as the effective volume source half-width plus the local plume spread, calculated at a given point as a weighted average over all upstream wakes;  $\sigma_{tot}$  is the combination of the upstream turbulent velocity and the local shear-induced turbulence:

$$\sigma_{tot} [i+1] = \sqrt{\sigma_{shear} [i]^2 + \sigma [i+1]^2}.$$
(10)

 $X_{crit}$  is a critical distance, which is the distance downstream from the effective source at which  $\sigma_{shear}$  starts to decay and is dependent on the inflow turbulence. For inflows with low turbulence, the shear-induced turbulence increases from zero at the

turbine to a maximum at  $X_{crit}$ , in proportion with the local wind speed deficit, and then decays; for turbulent inflows, the initial value of  $\sigma_{shear}$  is  $0.4|\Delta U|$  at the wind turbine and it decays immediately.  $X_{crit}$  is defined as

$$X_{crit} = \begin{cases} 4D & TI \leq TI_{lower} \\ 4D \left[ 1 - \frac{(TI - TI_{lower})}{(TI_{upper} - TI_{lower})} \right] & TI_{lower} < TI < TI_{upper}. \\ 0 & TI \geq TI_{upper} \end{cases}$$
(11)

The inflow turbulence is characterised by the turbulence intensity TI, which is expressed as a percentage and represents the ratio of the horizontal turbulence to the horizontal mean flow. TI is defined as

$$TI = 100 \times \frac{\sqrt{\sigma_u^2 + \sigma_v^2}}{\sqrt{u^2 + v^2}}.$$
(12)

$TI_{lower}$  and  $TI_{upper}$  are threshold values determined during validation of the model; these are set to 12% and 18% respectively.

# 2.4 Treatment of wind turbine interaction

Before the wake calculations are carried out for each input meteorological condition all the input wind turbines are reordered according to their downwind position. The assumption is made that wind turbines are only affected by the wakes

from upstream wind turbines. The flow field is modified by each wind turbine wake and includes the effect of the wakes from all upstream wind turbines.

The perturbation of the flow field by each wind turbine is represented by adjusting the values of the velocity components from the input flow field to include the calculated wind speed deficit. This is done over a grid of points covering the modelling region and the wind turbine locations at a range of heights within the atmospheric boundary layer. This process is

15 repeated iteratively to apply the effect of each wind turbine in downstream order. The change in the flow field for each wind turbine in a wind farm affects the characterisation of the effective volume source and the dispersion of the wake.

## 2.5 Marine boundary layer scheme

For offshore sites, FLOWSTAR-Energy includes a marine boundary layer scheme for calculating surface roughness and heat fluxes.

For surface roughness  $z_0$  the formula of Beljaars, 1994, has been adopted, as used by the European Centre for Medium-range Weather Forecasts (ECMWF):

$$z_0 = \alpha_m \frac{v}{u_*} + \alpha_{Ch} \frac{{u_*}^2}{g}$$
(13)

where  $u_*$  (ms<sup>-1</sup>) is the friction velocity, v (m<sup>2</sup>s<sup>-1</sup>) is the kinematic viscosity of air, g is the acceleration due to gravity (ms<sup>-2</sup>),  $\alpha_m = 0.11$  and  $\alpha_{Ch}$  is the Charnock parameter.  $\alpha_{Ch} = 0.08$  is typical for an offshore site not remote from the coast. Over the sea the surface roughness values for sensible heat ( $z_{0H}$ ) and moisture ( $z_{0q}$ ) are given by Beljaars , 1994, as

$$z_{0H} = \alpha_H \frac{v}{u_*} \text{ and } z_{0q} = \alpha_q \frac{v}{u_*}$$
(14)

where  $\alpha_H = 0.4$  and  $\alpha_q = 0.62$ .

The surface roughness for sensible heat is used to define velocity and concentration profiles, while the surface roughness for moisture is used to calculate the latent heat flux at the sea surface.

## **3 Validation**

The three datasets used for model validation are the Tjæreborg 60 m single wind turbine dataset; the NoordZee wind farm dataset and the Nysted wind farm dataset. These datasets were a key component of the EU's TOPFARM project (Larsen et al., 2011); a summary of the datasets is provided in Table 1. The definitions of the statistical measures used to compare modelled and observed data are given in Table 2.

#### 3.1 Tjæreborg 60 m Wind Turbine

- Wind speed measurements were recorded between 1988 and 1993 on two meteorological masts near to a wind turbine at Tjæreborg Enge wind farm in Esbjerg, Denmark, approximately 750 m downwind from the coast. The wind turbine hub height was 60 m and the rotor diameter (*D*) was 61 m. One of the masts, M1, was sited 122 m upstream of the turbine and the other mast, M2, was sited 122 m downstream of the turbine. The layout of the turbine and the measurement masts is shown in Figure 2. Wind speed measurements at mast M2 at heights 30, 45, 60 and 90 m have been categorised into four
- flow cases based on the turbine hub height wind speed at mast M1: 6, 8, 10 and  $12\pm0.5 \text{ ms}^{-1}$ . The data have also been categorised according to the wind direction, in 1 degree intervals up to 40 degrees either side of the case where mast M2 is directly downstream of the turbine. The thrust coefficient ( $C_T$ ) and power curves for this turbine are shown in Figure 3. The maximum theoretical value of  $C_T$  is 1, yet for wind speeds of 5, 6 and 7 ms<sup>-1</sup> the given value of  $C_T$  is greater than 1; therefore in the modelling analysis  $C_T = min(C_T, 1)$ .
- The modelling for this case assumed a constant surface roughness of 0.005 m; this is an appropriate value for the site, which is in flat, open grassland close to the sea. For each flow case, wind directions were modelled from 230 to 310 degrees inclusive, in 1 degree intervals with no wind direction sector averaging applied. In the absence of any additional meteorological data, or date and time information for the measurements, the modelling assumed neutral atmospheric stability.
- The results presented in Figure 4 show the modelled and observed wind speed deficit at mast M2, normalised by the wind speed at mast M1, for all four flow cases as a function of wind direction offset, where 0 degrees represents the case where the mast M2 is directly downstream of the turbine. The results are averaged over the measurement heights 30, 45, 60 and 90 m.

There is generally good agreement between the modelled and observed wake deficit, and the model simulates the shape of

30 the wake deficit well. However, while the modelled and measured data show broadly the same behaviour, namely that as the wind speed increases the strength of the wake decreases, this behaviour is less marked in the observed data than in the

modelled data. The modelled wake has a tendency to be slightly wider than the observed wake in this case. This is likely to be related to the assumption of neutral stability and also to the assumption of uniform surface roughness in this complex location