# Peer review of "FLOWSTAR-Energy: a high resolution wind farm wake model"

_Wind Energy Science, 2016_

## Referee Comment (RC1) · Anonymous Referee #1 · 14 Nov 2016

The manuscript attempts to describe how the flow model FLOWSTAR can be extended to account for wind farm wake. A wake deficit model is introduced which builds on concepts from the study of dispersion of plumes. A model for shear-induced (also refered to as wake-added) turbulence is also introduced. The manuscripts includes comparison with measured data from three sites, two of them offshore.

General comments:
1. There appears to be very little connection between FLOWSTAR and the super-imposed wake deficit model. The downstream propagation of the wakes appears for example to be independent of the the stream lines of the flow. I therefore recommend that the connection to FLOWSTAR is down-played until the validation in complex terrain is in place.

2. The manuscript does not include validation of the calculated shear-induced (wake-added) turbulence. The connection between the model for shear-induced turbulence and wake deficit model appears to be one-way, such that the shear-induced turbulence model is not necessary for the description of the wake deficit. I therefore recommend that the description of the shear-induced turbulence model is removed until the output of this model is validated. Influencing this recommendation is the observation that many of equations in section 2.3 have generated specific or technical comments below.

Specific comments:
1. Page 5, line 13: In eq. (8) it is unclear if $\sigma_{shear}$ is the turbulence contribution from one WTG or the accumulation of the wake-added turbulence from all upstream WTG:s. The formula seems to suggest the former.
2. Page 5, line 22: What is $\sigma[i+1]$ and how is it calculated?
3. Page 6, line 4: Eq. (12) leads to a higher value of TI than the standard definition. Is the introduction of a non-standard definition intentional?

Technical comments:
1. Page 3, line 24: Reference is missing.
2. Page 4, line 25: A "crosswind vertical slice" is not an intuitive concept. Consider adding a sketch to illustrate the concept.
3. Page 5 line 1: Eq. (7) appears to be missing a $y_s$ (corresponding to $z_s$).
4. Page 5, line 19: In eq. (9) $\sigma_{tot}$ should be $\sigma_{tot}[i]$ or $\sigma_{tot}[i+1]$?

---

## Referee Comment (RC2) · C. Masson (Referee) · 16 Nov 2016

General comments: This is a well written paper and the results presented are quite convincing in term of power. It is worth publishing once the following comments are treated properly. For example, it is unclear to the reviewer how to get the 'complete' flow field in the wakes (for example how to get results from Figs. 9 and 14). Is it my limit of understanding or the way the methodology is described/presented? I have read several times the paper and some of the references and I do not fully understand how the wakes are introduced in the flow using FLOWSTAR. However, as mentioned before, the results are convincing. So I think rewriting the methodology is necessary. Also, equation (7) is presented with no explanation how to get it and/or relevant references. It might be classical information for researchers at CERC but sufficient information should be provided to the readers. Complete information regarding the calculation of

some parameters in equation (7) are missing. This makes the duplication and/or the verification of the results impossible. I wanted to mention this point but will leave the journal editor dealing with this aspect. In the validation section for the Tjaereborg 60m wind turbine, the results are averaged over the measurement heights (see page 7, line 27) without any discussion/justification. I would be curious to know why it is averaged. Regarding the wind farm results, they are all obtained for off-shore setting. This is not complex surfaces justifying the use of FLOWSTAR. This is clearly mentioned at the end of the abstract and conclusion. I have appreciated to see this clearly stated. Nevertheless, why presenting FLOWSTAR then?

Specific comments:

1. Please remove 'FLOWSTAR-Energy' from the title. It is not really necessary.

2. Page 2, lines 26-28: there are errors in referencing.

3. Page 3, line 24: Following 0 ????

---

## Referee Comment (RC3) · Anonymous Referee #3 · 25 Nov 2016

Review of Amy Stidworthy % David Carruthers: FLOWSTAR-Energy: a high resolution wind farm wake model

The paper describes a wind farm wake model that essentially treats the momentum deficit caused by rotor drag as passive tracer. This is not a new idea, and it might even be a good one.

I have several concerns about the results in this paper:

The whole set of model equations is not presented, and proper references are not give. There should be references, including equation numbers, to each and every model eqauation so that the (enthusiastic) reader would be able to recreate the calculations. There should also be a table of values of numerical constants used. Otherwise the model is just a black box of little interest except to its owner.

The 'typical value' of the Charnock constant is extreme.

The source term for the momentum deficit is 3 times too high. Equation (6) is simply stated witout any attempt to argue for it, and unfortunately it is wrong.

The origin of (8) is a mystery, except that is should somehow be 'based' on a wind tunnel experiment with laminar inflow. The factor 0.4 appearing in (8), indicating a large influence of shear on the turbulence, appears out of the blue without explanation. WT wake measurements indicate an enhancement of turbulence in the wake combined with a reduction of the turbulent length scale so that the turbulent diffusivity does not change that much.

The constants TIupper and TIlower have been 'determined during validation of the model'. This is strictly forbidden.

The fractional bias (Nysted data) is miraculously close to zero given the fact that the momentum source term has been set a factor of 3 too high. I can't help speculating whether this may have been achieved by tweaking the Charnock constant and perhaps other constants. There is nothing in the text that can make me think otherwise.

I recommend not to publish the paper in its present form. Perhaps after substantial revision, but in any case the calculations have to be done again using the correct source term. This will probably change the model results substantially and it is difficult to guess the impact on conclusions.

Comments on the fly (while reading):

p. 2 'Error! Reference source not found'. Twice

My library does not have the CERC reports referred to in section 2 and I could not find them on the net, not even on the CERC web site. I have had troubbles finding other references too, such as Carruthers 1988. This is serious, because model asumptions are only explained very rudimentally in the your paper. I am missing a concise explanation of what your model is all about.

p. 3

line 17 You say that the dispersion of the wake from a given turbine is influenced by the wakes of upstream turbines. How exactly? It sounds as if you are not treating the momentum deficit as a passive tracer after all.

According to Hansen ac=0.4, not 0.2. This gives a critical Ct of 0.96 instead of 0.64 (in eq. 2). Wind turbine Ct values as high as 0.96 are rare, but Ct>0.64 occurs often. It therefore matters if you set ac as low as 0.2. Is there any experimental evidence for ac=0.2? Why not simply use measured values?

p. 4 The source is a square disk, that has a volume! But ok, fig. 1 explains it.

The correct source strength must be

Q Vsrc = Thrust/rho = $\frac{1}{2}$ Ct V^2 pi R^2

and, since Vsrc=dx pi R^2(1-a)/(1-2a), I get

Q=2a(1-2a)V^2/dx

This differs from (6) by a factor 1-2a. Taking the typical value a=1/3, 1-2a=1/3 so that you get 3 times larger Q than I do. I think the reason is that you the advection speed at the 'virtual' source as V instead of (1-2a)V. It is true that the dispersion model does not see any reduction of advection speed, but we have not begun to disperse anything yet. In other words, first Q should determined so that it is consistent with the thrust, and then we decide what wake model to use to disperse the momentum deficit. This is quite serious, a factor of 3 will of course completely change the results.

Who is 'the receptor'?

p. 5

What is the sign of the reflection term in (7) and why? ADMS uses non-Gaussian plumes in unstable conditions. Has the been drooped in your model?
[Figure]

You don't give many detailes about dispersion model, and the references (Hunt 1899, Hanna 1989, Weil 1985) do not seem to adress the ADMS model. You need to give a reference that contains the exact equation that are using. It would have been nice if you had presented the whole model here, and I understand that it would perhaps be a too long story. Onthe other hand, 14 lines is perhaps too short. I suggest you add a short description of how the dispersion parameters are determined from the turbulence and the need to take turbulence generation by wake shear into account.

Section 2.3 presents formulas based on Bevilaqua and Lykoudis (1978), but they cannot be found in the reference. Where do they come from?

B&L used an essentially laminar wind tunnel with Ti<0.3% in the inflow. What makes their results relevant for wakes with turbulent inflow?

p. 6 In (12) '100' should be deleted. If you insist, you could write '100%' here, which of course is equal to one 1.

Line 5: "Tilower and Tiupper are threshold values determined during validation of the model". Tweeking model constants during validation is not allowed. It invalidates the 'validation' and it is not acceptable.

A Charnoch parameter of 0.08 is extreme rather than typical. 0.01 to 0.02 is typical.

p. 7

What role does humidity play in the model?

What exactly is it that is located 750m downwind from the coast?

You limit Ct to 1, so (1) was not used after all. Ct>1 in fig.3. Confusing. In fig. 3 I take it that Ct was made from using (1). Where does the power curve come from?

1 degree wide bins are dangerous because 'the' wind direction cannot be known with that precision. Two different wind vanes will produce two different 10 minutes averages, often deviating several degrees. Successive ten minutes averages differ typically by

about 5 degrees, and there can be large scale spatial inhomogenities. As a result many models will predict too large wake effects for narrow wind direction bins centrered around a direction aligned with a WT row. Predictions for wide wd bins, which are less sensitive to wd ubcertainties, work much better. You may claim the your model is based on measurements and therefore the wind direction uncertainty is built into sigma_y. In that case your results should be ok for both narrow and wide bins. You should check this.

p. 8

Results for the 5 degree wd bin should be supplemented by results from wider bins. Ok, you do it for the Nysted data.

Why is the power from a turbine used to obtain the windspeed instead of the measurements at the met mast?

Section 0???

It is inconsistent to assuming neutral conditions when calculating z0, and then feed the model with very unstable conditions.

I cannot reproduce the roughnesses listed in table 4.

Where do the stability distributions in table 3 come from (what measurements)? They don't immediately seem to be very realistic.

The relevant error bar is the standard error = the standard deviation of the estimated mean value = standard deviation/sqrt(#observations).

p. 9

How was LMO measured at Nysted? With a sonic?

p. 10

Section 0 again.

Both power production and probability vary across a 1m/s wind speed bin which can affect the result as you say. It is probably better to take averages of the ratio of the turbine production and production of the reference turbine(s).

---

## Author Comment (AC1) · 20 Dec 2016

(Reviewer comments are labelled 'RC' and numbered with the reviewer number and the comment number; the author comments are labelled 'AC' and numbered the same way. AC1.1 is the authors' response to reviewer comment RC1.1 (the first comment from reviewer 1) etc.)

RC1.1: "There appears to be very little connection between FLOWSTAR and the superimposed wake deficit model. The downstream propagation of the wakes appears for example to be independent of the stream lines of the flow. I therefore recommend that the connection to FLOWSTAR is down-played until the validation in complex terrain is in place."

AC1.1: The wake model and the FLOWSTAR flow model are integrated, in the same

way that the plume model is integrated with FLOWSTAR in ADMS. The centreline of the wake does follow the streamlines of the flow; however this is not adequately explained in the paper and so amendments will be made.

RC1.2: "The manuscript does not include validation of the calculated shear-induced (wake added) turbulence. The connection between the model for shear-induced turbulence and wake deficit model appears to be one-way, such that the shear-induced turbulence model is not necessary for the description of the wake deficit. I therefore recommend that the description of the shear-induced turbulence model is removed until the output of this model is validated. Influencing this recommendation is the observation that many of equations in section 2.3 have generated specific or technical comments below."

AC1.2: Although the authors agree with the reviewer that the shear-induced turbulence component has not been explicitly validated in this paper, we disagree with the reviewer's statement that the description of the shear-induced turbulence model should be removed from this paper. The shear-induced turbulence in the wake model affects the dispersion of individual wakes, and impacts on the source characterisation and dispersion of downstream wakes; therefore the turbulence model is implicitly validated in the Nysted and Noordzee wind farm validation cases and to remove the description of the shear-induced turbulence model would render the model description incomplete. However, we acknowledge that this linkage is not adequately explained in the paper so amendments will be made to rectify this.

RC1.3: p5, line 13 "In eq. (8) it is unclear if sigma-shear is the turbulence contribution from one WTG or the accumulation of the wake-added turbulence from all upstream WTGs. The formula seems to suggest the former."

AC1.3: Yes, sigma-shear is the turbulence from one WTG; the text will be amended to explain this better.

RC1.4: p5, line 22 "What is sigma[i + 1] and how is it calculated?"

AC1.4: sigma[i+1] is the total turbulence at the [i+1]th point, not including shear-induced turbulence from the current turbine. It does therefore include ambient turbulence and the shear-induced turbulence generated by upstream turbines. Again, the text will be amended to explain this.

RC1.5: p6, line 4 "Eq. (12) leads to a higher value of TI than the standard definition. Is the introduction of a non-standard definition intentional?"

AC1.5: Yes. The definition of turbulence intensity used for the calculation of shear induced turbulence accounts for the spatially-varying nature of the wind direction and the potential influence of complex terrain.

RC1.6: p3, line 24 "Reference is missing." AC1.6: Apologies, errors with links were introduced when the paper was re-formatted for Wind Energy Science; all links and references will be re-checked and corrected prior to re-submission.

RC1.7: p4, line 25 "A 'crosswind vertical slice' is not an intuitive concept. Consider adding a sketch to illustrate the concept."

AC1.7: Agreed, a sketch will be added. This section will be revised significantly in the light of comments from all three reviewers, so hopefully this will help clarify the methodology.

RC1.8: p5, line 1 "Eq. (7) appears to be missing a ys (corresponding to zs)"

AC1.8: (x,y,z) is a ground-based source-centred coordinate system, i.e. x=0 and y=0 at the source location, hence equation (7) is correct. However, this is not explained in the text. This section will be revised significantly, and an explanation of the coordinate system will be included.

RC1.9: p5, line 19 "In eq. (9) sigma-tot should be sigma-tot[i] or sigma-tot[i+1]?"

AC1.9: It should be sigma-tot[i]. This will be corrected.

---

## Author Comment (AC2) · 20 Dec 2016

(Reviewer comments are labelled 'RC' and numbered with the reviewer number and the comment number; the author comments are labelled 'AC' and numbered the same way. AC2.1 is the authors' response to reviewer comment RC2.1 (the first comment from reviewer 2) etc.)

RC2.1: "It is unclear to the reviewer how to get the 'complete' flow field in the wakes (for example how to get results from Figs. 9 and 14). Is it my limit of understanding or the way the methodology is described/presented? I have read several times the paper and some of the references and I do not fully understand how the wakes are introduced in the flow using FLOWSTAR."

AC2.1: Earlier drafts of the paper included a more complete description of the plume

model that underlies the wake model, but this section was removed to shorten the paper, in the belief that the referenced papers gave sufficient information on this aspect; however, from the comments from all three reviewers, it is clear that there is insufficient information in the paper on this aspect, so this section will be reinstated.

RC2.2: "Equation (7) is presented with no explanation how to get it and/or relevant references. It might be classical information for researchers at CERC but sufficient information should be provided to the readers. Complete information regarding the calculation of some parameters in equation (7) are missing. This makes the duplication and/or the verification of the results impossible."

AC2.2: The authors acknowledge that there is an issue here, which will be addressed by the inclusion of a fuller description of the plume model that underlies the wake model.

RC2.3: "In the validation section for the Tjaereborg 60m wind turbine, the results are averaged over the measurement heights (see page 7, line 27) without any discussion/justification. I would be curious to know why it is averaged."

AC2.3: The Tjaereborg results were averaged over the measurement heights in order to simplify the presentation of results, presenting 4 graphs instead of 16; however, all 16 graphs can be presented if this is preferred.

RC2.4: "Regarding the wind farm results, they are all obtained for off-shore setting. This is not complex surfaces justifying the use of FLOWSTAR. This is clearly mentioned at the end of the abstract and conclusion. I have appreciated to see this clearly stated. Nevertheless, why presenting FLOWSTAR then?"

AC2.4: The wake model is integrated with FLOWSTAR in FLOWSTAR-Energy; this first FLOWSTAR-Energy paper addresses the performance of the wake model in flat terrain and offshore. A subsequent paper will present validation of the wake model in complex terrain, as is noted in the abstract and in the conclusion.

RC2.5: "Please remove 'FLOWSTAR-Energy' from the title. It is not really necessary."

AC2.5: The title can be changed if this is thought necessary. However it is common to include model names in titles.

RC2.6: "Page 2, lines 26-28: there are errors in referencing."

AC2.6: Apologies, errors with links were introduced when the paper was re-formatted for Wind Energy Science; all links and references will be re-checked and corrected prior to re-submission.

RC2.7: "Page 3, line 24: Following 0 ????"

AC2.7: Apologies, errors with links were introduced when the paper was re-formatted for Wind Energy Science; all links and references will be re-checked and corrected prior to re-submission.

---

## Author Comment (AC3) · 20 Dec 2016

(Reviewer comments are labelled 'RC' and numbered with the reviewer number and the comment number; the author comments are labelled 'AC' and numbered the same way. AC3.1 is the authors' response to reviewer comment RC3.1 (the first comment from reviewer 3) etc.)

RC3.1: "The whole set of model equations is not presented, and proper references are not give. There should be references, including equation numbers, to each and every model eqauation so that the (enthusiastic) reader would be able to recreate the calculations. There should also be a table of values of numerical constants used. Otherwise the model is just a black box of little interest except to its owner."

AC3.1: We acknowledge that the paper requires a fuller description of the ADMS plume

model underlying the wake model, and how it is integrated with the FLOWSTAR flow model. Previous drafts of the paper included a section on this aspect, but this was removed prior to submission to shorten the paper, as the authors felt there was sufficient information in the referenced papers. However, based on the comments of all three reviewers, this section clearly needs to be reinstated.

RC3.2: "The 'typical value' of the Charnock constant is extreme."

AC3.2: This is not the case. While the Charnock constant value used (0.08) is higher than the typical values in the literature for open marine sites far from land, validation work done at CERC in 2004, as part of a project for the UK Government's Department for Trade and Industry (DTI), showed that for offshore sites where the immediate fetch is sea, but where land is nearby, higher values of the Charnock constant were required for the ECMWF scheme to calculate wind speeds accurately. This is written up in a report accessible on the CERC website (link below); the above justification and reference will be added to the paper. We also refer the reviewer to the values of z0 we calculate (Tables 4 and 6). http://www.cerc.co.uk/environmental-research/assets/data/CERC_2004_DTI_Development_of_boundary_layer_profiles.pdf.

RC3.3: "The source term for the momentum deficit is 3 times too high. Equation (6) is simply stated witout any attempt to argue for it, and unfortunately it is wrong."

AC3.3: The source term given in Equation (6) is correct in the paper; it is consistent with the methodology in the FLOWSTAR-Energy wake model. The FLOWSTAR-Energy wake model treats an individual wake as a plume of material, with concentration used as a surrogate for wind speed deficit, and does not adjust the flow speed within the wake being modelled; hence, the use of the inflow wind speed U in the calculation of the source strength gives the required initial, maximum, theoretical, wind speed deficit. After all, the use of a volume source at all is just an artificial construct, designed to give the required wind speed deficit in the required location. If the model did adjust the flow speed in the wake, then yes it would be appropriate to apply the factor (1-
2a) in the calculation of the source strength, but because it doesn't, it isn't. Once the wake calculations have been completed for an individual wake, the 3D perturbation to the flow field from the resulting wind speed deficit is applied to the 'ambient' flow field before the wake calculations for the next downstream turbine are carried out; hence the wind speed deficit (and added turbulence) due to each wake does have an effect on the source characterisation for all downstream wakes. More explanation on this aspect will be added to the paper.

RC3.4: "The origin of (8) is a mystery, except that is should somehow be 'based' on a wind tunnel experiment with laminar inflow. The factor 0.4 appearing in (8), indicating a large influence of shear on the turbulence, appears out of the blue without explanation. WT wake measurements indicate an enhancement of turbulence in the wake combined with a reduction of the turbulent length scale so that the turbulent diffusivity does not change that much."

AC3.4: We agree that this needs further explanation in the paper. The approach is to generate additional turbulence from breakdown of the vortex tube. This takes time to occur when the turbulence is weak (downstream of the first turbine in stable flows), but contributes immediately in more turbulent (neutral/convective) flows where its relative contribution is much lower.

RC3.5: "The constants TIupper and TIlower have been 'determined during validation of the model'. This is strictly forbidden."

AC3.5: Again, we agree that this needs further explanation in the paper. The constants TIupper and TIlower were refined during the Tjaereborg single-turbine validation work, not the Nysted and Noordzee wind farm assessments. They represent the different inflow stability regimes: TI<=12% represents very stable flow, 12%<TI<=18% represents stable/neutral flow and TI>18% represents unstable flow. Note that the definition of TI used in equation (8) is the ratio of the horizontal turbulence to the horizontal flow (TI=sqrt(sigma-u**2+sigma-v**2)/U), not the more standard definition that uses only

the longitudinal component of the turbulence (TI=sigma-u/U); the values of TI using this definition are higher than those calculated using the standard definition. We use this non-standard definition to account for the spatially-varying nature of the wind direction and the potential influence of complex terrain.

RC3.6: "The fractional bias (Nysted data) is miraculously close to zero given the fact that the momentum source term has been set a factor of 3 too high. I can't help speculating whether this may have been achieved by tweaking the Charnock constant and perhaps other constants. There is nothing in the text that can make me think otherwise."

AC3.6: The source term is correct for FLOWSTAR-Energy (as explained above), and the value of the Charnock parameter used has already been justified; no other constants have been changed to achieve these results.

RC3.7: p2 "'Error! Reference source not found'. Twice"

AC3.7: Apologies, errors with links were introduced when the paper was re-formatted for Wind Energy Science; all links and references will be re-checked and corrected prior to re-submission.

RC3.8: "My library does not have the CERC reports referred to in section 2 and I could not find them on the net, not even on the CERC web site. I have had troubbles finding other references too, such as Carruthers 1988. This is serious, because model asumptions are only explained very rudimentaly in the your paper. I am missing a concise explanation of what your model is all about."

AC3.8: This is unfortunate, because the editor previously contacted the authors to request electronic copies of these papers for the other 2 reviewers, and the authors sent them without delay. These papers can of course also be made available to this third reviewer, but also the additional section that will be added about volume source dispersion (discussed above) should help clarify the methodology.

RC3.9: "p3, line 17: You say that the dispersion of the wake from a given turbine is influenced by the wakes of upstream turbines. How exactly? It sounds as if you are not treating the momentum deficit as a passive tracer after all."

AC3.9: Once the wake calculations have been completed for an individual wake, the 3D perturbation to the flow field from the resulting wind speed deficit and turbulence field is applied to the 'ambient' flow field before the wake calculations for the next downstream turbine are carried out; hence the wind speed deficit (and added turbulence) due to each wake does have an effect on the source characterisation and dispersion of all downstream wakes. More explanation on this aspect will be added to the paper.

RC3.10: "p3, line 17: According to Hansen ac=0.4, not 0.2. This gives a critical Ct of 0.96 instead of 0.64 (in eq. 2). Wind turbine Ct values as high as 0.96 are rare, but Ct>0.64 occurs often. It therefore matters if you set ac as low as 0.2. Is there any experimental evidence for ac=0.2? Why not simply use measured values?"

AC3.10: We disagree; according to Hansen, page 53, equation (6.38), ac is approximately 0.2, as stated in the paper.

RC3.11: "p4 The correct source strength must be Q Vsrc = Thrust/rho = Ct V^2 pi R^2 and, since Vsrc=dx pi R^2(1-a)/(1-2a), I get Q=2a(1-2a)V^2/dx This differs from (6) by a factor 1-2a. Taking the typical value a=1/3, 1-2a=1/3 so that you get 3 times larger Q than I do. I think the reason is that you the advection speed at the 'virtual' source as V instead of (1-2a)V. It is true that the dispersion model does not see any reduction of advection speed, but we have not begun to disperse anything yet. In other words, first Q should determined so that it is consistent with the thrust, and then we decide what wake model to use to disperse the momentum deficit. This is quite serious, a factor of 3 will of course completely change the results."

AC3.11: The source term given in Equation (6) is correct in the paper; it is consistent with the methodology in the FLOWSTAR-Energy wake model. The FLOWSTAR-Energy wake model treats an individual wake as a plume of material, with concentration used

as a surrogate for wind speed deficit, and does not adjust the flow speed within the wake being modelled; hence, the use of the inflow wind speed U in the calculation of the source strength gives the required initial, maximum, theoretical, wind speed deficit. After all, the use of a volume source at all is just an artificial construct, designed to give the required wind speed deficit in the required location. If the model did adjust the flow speed in the wake, then yes it would be appropriate to apply the factor (1-2a) in the calculation of the source strength, but because it doesn't, it isn't. Once the wake calculations have been completed for an individual wake, the 3D perturbation to the flow field from the resulting wind speed deficit is applied to the 'ambient' flow field before the wake calculations before the next downstream turbine are carried out; hence the wind speed deficit (and turbulence) due to each wake does have an effect on the source characterisation for all downstream wakes. More explanation on this aspect will be added to the paper.

RC3.12: "p4: Who is 'the receptor'?"

AC3.12: The 'receptor' is the output location in question. This section will be revised significantly before re-submission to more clearly explain the ADMS plume model that underlies the wake model.

RC3.13: "p5: What is the sign of the reflection term in (7) and why? ADMS uses non-Gaussian plumes in unstable conditions. Has the been drooped in your model?"

AC3.13: No, the non-Gaussian plume in unstable conditions has not been dropped; it is part of FLOWSTAR-Energy; this section will become clearer when this section is revised in line with the comments above.

RC3.14: "p5: You don't give many detailes about dispersion model, and the references (Hunt 1899, Hanna 1989, Weil 1985) do not seem to adress the ADMS model. You need to give a reference that contains the exact equation that are using. It would have been nice if you had presented the whole model here, and I understand that it would perhaps be a too long story. Onthe other hand, 14 lines is perhaps too short. I suggest

you add a short description of how the dispersion parameters are determined from the turbulence and the need to take turbulence generation by wake shear into account."

AC3.14: We fully appreciate this point. As already mentioned above, previous drafts of the paper included a section on this aspect, but this was removed prior to submission to shorten the paper, as the authors felt there was sufficient information in the referenced papers. However, based on the comments of all three reviewers, this section clearly needs to be reinstated.

RC3.15: "p5: Section 2.3 presents formulas based on Bevilaqua and Lykoudis (1978), but they cannot be found in the reference. Where do they come from?"

AC3.15: These equations have been developed at CERC, based on Bevilaqua and Lykoudis, to express the shear-induced turbulence component applied in the wake model. The approach is to generate additional turbulence from breakdown of the vortex tube. This takes time to occur when the turbulence is weak (downstream of the first turbine in stable flows), but contributes immediately in more turbulent (neutral/convective) flows where its relative contribution is much lower. This will be explained further in the paper.

RC3.16: "p5: B&L used an essentially laminar wind tunnel with Ti<0.3% in the inflow. What makes their results relevant for wakes with turbulent inflow?"

AC3.16: The authors have used the principles described in the B&L reference to develop a model of shear-induced turbulence as described in AC3.15 above. This will be explained further in the paper.

RC3.17: "p. 6: In (12) '100' should be deleted. If you insist, you could write '100%' here, which of course is equal to one 1."

AC3.17: Noted - this will be changed.

RC3.18: "p6, line 5: 'Tilower and Tiupper are threshold values determined during validation of the model'. Tweeking model constants during validation is not allowed. It invalidates the 'validation' and it is not acceptable."

AC3.18: This needs further explanation in the paper. The constants TIupper and TIlower were refined during the Tjaereborg single-turbine validation work, not the Nysted and Noordzee wind farm assessments. They represent the different inflow stability regimes: TI<=12% represents very stable flow, 12%<TI<=18% represents stable/neutral flow and TI>18% represents unstable flow. Note that the definition of TI used in equation (8) is the ratio of the horizontal turbulence to the horizontal flow (TI=sqrt(sigma-u**2+sigma-v**2)/U), not the more standard definition that uses only the longitudinal component of the turbulence (TI=sigma-u/U); the values of TI using this definition are higher than those calculated using the standard definition. We use this non-standard definition to account for the spatially-varying nature of the wind direction and the potential influence of complex terrain.

RC3.19: "p6: A Charnoch parameter of 0.08 is extreme rather than typical. 0.01 to 0.02 is typical."

AC3.19: This is not the case. While the Charnock constant value used (0.08) is higher than the typical values in the literature for open marine sites far from land, validation work done at CERC in 2004, as part of a project for the UK Government's Department for Trade and Industry (DTI), showed that for offshore sites where the immediate fetch is sea, but where land is nearby, higher values of the Charnock constant were required for the ECMWF scheme to calculate wind speeds accurately. This is written up in a report accessible on the CERC website (link below); the above justification and reference will be added to the paper. We could also refer the reviewer to the values of z0 we calculate which are reasonable. http://www.cerc.co.uk/environmental-research/assets/data/CERC_2004_DTI_Development_of_boundary_layer_profiles.pdf.

RC3.20: "p7: What role does humidity play in the model?"

AC3.20: Humidity does not play any role in FLOWSTAR-Energy

RC3.21: "p7: What exactly is it that is located 750m downwind from the coast?"

AC3.21: The Tjaereborg wind turbine is 750 m downwind from the coast. This will be clarified in the text.

RC3.22: "p7: You limit Ct to 1, so (1) was not used after all. Ct>1 in fig.3. Confusing. In fig. 3 I take it that Ct was made from using (1). Where does the power curve come from?"

AC3.22: Ct values greater than 1 do not make physical sense, which is why we limit Ct to be <= 1; the definition of (1) will be amended to make this limit clear. Fig 3 was made using power and Ct data provided to the authors during the TOPFARM project by DTU. This is stated in the acknowledgements at the end of the paper, but could be stated more explicitly in the section for each validation case.

RC3.23: "p7: 1 degree wide bins are dangerous because 'the' wind direction cannot be known with that precision. Two different wind vanes will produce two different 10 minutes averages, often deviating several degrees. Successive ten minutes averages differ typically by about 5 degrees, and there can be large scale spatial inhomogenities. As a result many models will predict too large wake effects for narrow wind direction bins centrered around a direction aligned with a WT row. Predictions for wide wd bins, which are less sensitive to wd ubcertainties, work much better. You may claim the your model is based on measurements and therefore the wind direction uncertainty is built into sigma_y. In that case your results should be ok for both narrow and wide bins. You should check this."

AC3.23: The authors agree completely with the reviewer here, but the data for Tjaereborg were provided in 1 degree wide bins, which is why the modelling for Tjaereborg has also used 1 degree wide bins. The Noordzee case uses a 5 degree bin (again, this is how the data were supplied); the Nysted case uses a 30-degree bin (here, the data were supplied in 5 degree bins and have been aggregated into a 30-degree bin).

RC3.24: "p8: Results for the 5 degree wd bin should be supplemented by results from wider bins."

AC3.24: The data for the Noordzee case was supplied for a single 5-degree wind direction bin; the data for the Nysted case was supplied in multiple 5-degree bins and has been aggregated into a single 30-degree bin.

RC3.25: "p8: Why is the power from a turbine used to obtain the windspeed instead of the measure- ments at the met mast?"

AC3.25: The power from a reference turbine has been used to obtain the wind speed instead of the met mast measurements because the wind direction chosen for analysis is one where the met mast is affected by turbine wakes (see fig 6). The text will be amended to explain this.

RC3.26: "p8: Section 0???"

AC3.26: Apologies, errors with links were introduced when the paper was re-formatted for Wind Energy Science; all links and references will be re-checked and corrected prior to re-submission.

RC3.27: "p8: It is inconsistent to assuming neutral conditions when calculating z0, and then feed the model with very unstable conditions."

AC3.27: Noted, but there were no sea surface temperature data available, so an as- sumption of neutral stability had to be made in the marine boundary layer scheme. This is already stated in the text.

RC3.28: "p8: I cannot reproduce the roughnesses listed in table 4."

AC3.28: Equation (13) is solved iteratively in the model for z0, since u* depends on z0. However, we acknowledge the paper does not explain this, so it will be amended to explain this.

RC3.29: "p8: Where do the stability distributions in table 3 come from (what measure- ments)? They don't immediately seem to be very realistic." AC3.29: Again, these data were provided by DTU as part of the TOPFARM project; the answer to this question is

not contained in the dataset, so we will ask DTU for this information.

RC3.30: "p8: The relevant error bar is the standard error = the standard deviation of the estimated mean value = standard deviation/sqrt(#observations)."

AC3.30: Acknowledged; we will re-create the graphs so that the error bars represent the standard error rather than the standard deviation.

RC3.31: "p9: How was LMO measured at Nysted? With a sonic?"

AC3.31: Again, the answer to this question is not contained in the dataset, so we will ask DTU for this information.

RC3.32: "p10: Section 0 again"

AC3.32: Apologies, errors with links were introduced when the paper was re-formatted for Wind Energy Science; all links and references will be re-checked and corrected prior to re-submission.

RC3.33: "p10: Both power production and probability vary across a 1m/s wind speed bin which can affect the result as you say. It is probably better to take averages of the ratio of the turbine production and production of the reference turbine(s)."

AC3.33: Possibly, but plots of power, rather than normalised power, give a more complete picture of model behaviour compared with measurements.